# Evidence for a Strong Relationship between the Cytotoxicity and Intracellular Location of β-Amyloid

**DOI:** 10.3390/life12040577

**Published:** 2022-04-13

**Authors:** Md. Aminul Haque, Md. Selim Hossain, Tahmina Bilkis, Md. Imamul Islam, Il-Seon Park

**Affiliations:** 1Department of Biomedical Sciences, Chosun University, Gwangju 61452, Korea; aminul.haque@bracu.ac.bd (M.A.H.); selim@chosun.kr (M.S.H.); bilkis.tahmina20@cvasu.ac.bd (T.B.); md.islam3@umanitoba.ca (M.I.I.); 2Department of Cellular and Molecular Medicine, Chosun University, Gwangju 61452, Korea

**Keywords:** β-amyloid, Alzheimer’s disease, cytotoxicity, cell permeability, oligomeric species, tAβ42

## Abstract

β-Amyloid (Aβ) is a hallmark peptide of Alzheimer’s disease (AD). Herein, we explored the mechanism underlying the cytotoxicity of this peptide. Double treatment with oligomeric 42-amino-acid Aβ (Aβ42) species, which are more cytotoxic than other conformers such as monomers and fibrils, resulted in increased cytotoxicity. Under this treatment condition, an increase in intracellular localization of the peptide was observed, which indicated that the peptide administered extracellularly entered the cells. The cell-permeable peptide TAT-tagged Aβ42 (tAβ42), which was newly prepared for the study and found to be highly cell-permeable and soluble, induced Aβ-specific lamin protein cleavage, caspase-3/7-like DEVDase activation, and high cytotoxicity (5–10-fold higher than that induced by the wild-type oligomeric preparations). Oligomeric species enrichment and double treatment were not necessary for enhancing the cytotoxicity and intracellular location of the fusion peptide. Taiwaniaflavone, an inhibitor of the cytotoxicity of wild-type Aβ42 and tAβ42, strongly blocked the internalization of the peptides into the cells. These data imply a strong relationship between the cytotoxicity and intracellular location of the Aβ peptide. Based on these results, we suggest that agents that can reduce the cell permeability of Aβ42 are potential AD therapeutics.

## 1. Introduction

β-Amyloid (Aβ) is a group of 39–43 amino acid-long peptides [1] that are generated through the proteolytic cleavage of amyloid precursor proteins by α-, β-, and γ-secretases [2]. The conformation of Aβ peptides spontaneously transforms into unstable β-sheet-rich intermediate structures that interact with each other to form aggregates, such as oligomers, protofibrils, and fibrils [3]. The conformational species of Aβ plays a major role in the cytotoxicity of this peptide and is an important subject for understanding the mechanism underlying its cytotoxicity. Deposition of fibrillar aggregates of Aβ in the brain parenchyma and cerebral blood vessels [4] is thought to be the main cause of neurodegeneration in Alzheimer’s disease (AD) [5]. However, many reports have indicated that soluble Aβ oligomers and protofibrils are more cytotoxic than other structural species, such as Aβ fibrils, implying that these species are the primary cytotoxic factors underlying the development of AD [6,7,8,9,10,11].

One of the key issues regarding the Aβ peptide is determining the characteristics that are associated with its cytotoxicity. The oligomeric conformer of the peptide is a reasonable target for investigation, as it shows strong cytotoxicity, as mentioned previously. Aβ peptide species can exert toxic effects outside the cells. For instance, it has been already reported that the human leukocyte immunoglobulin-like receptor B2, cellular prion protein, and Fcγ receptor IIb are examples that have an affinity for Aβ oligomer and contributes to human AD neuropathology [12,13]. Thus, it is hypothesized that the solubility, and probably the structural characteristics, of the oligomeric species lead to cell death in the extracellular environment.

Moreover, it was proposed that the accumulation of the intraneuronal 42-amino-acid Aβ (Aβ42) is a key event in the neurodegenerative process. A related report showed that long-term expression of human amyloid precursor protein (APP) in rat cortical neurons induces apoptosis [14]. The same study reported that the extracellular Aβ40 produced via APP processing did not induce neural death, whereas intracellular Aβ42 accumulated via the expression of full-length APP led to apoptosis in neurons. Considering that the oligomeric species of Aβ could be located intracellularly by endocytosis [15], and less toxic species such as fibrils are not [16], it is also possible that the intracellular location of the oligomers makes the peptide highly cytotoxic.

Although both intra- and extracellular Aβ peptides can cause cytotoxicity, peptide location can influence the level of cytotoxicity. If the cytotoxic process based on the latter hypothesis is more influential in determining the level of cytotoxicity, the cell permeability of the peptide could be a crucial factor in determining the level of cytotoxicity. In the current study, we explored a possible association between the intracellular location and cytotoxicity of Aβ peptides and provided evidence for a strong correlation between them.

## 2. Materials and Methods

### 2.1. Materials

Taiwaniaflavone was isolated as previously described [17]. Maltose binding protein (MBP) was purified as previously described [17,18]. All other chemicals were procured from Sigma-Aldrich (St. Louis, MO, USA) unless otherwise indicated.

### 2.2. Purification and Preparation of Aβ42 and tAβ42 Peptides

The Aβ42 and a reverse form of Aβ42 (r-Aβ42), were purified and prepared using a previously constructed pET28b-GroES-ubiquitin-Aβ42 and pET28b-GroES-ubiquitin-Aβ(42-1), as described in a previous study [19]. tAβ42, an Aβ42 peptide harboring cell-permeable peptide TAT [20,21] in its N-terminus, was produced using pET28-GroES-ubiquitin-TAT-Aβ42 which was constructed via the insertion of TAT sequences into the above Aβ42 plasmid using a DpnI-mediated single-step site-directed mutagenesis kit (New England Biolabs, UK). The sense and antisense primers for the polymerase chain reaction (PCR) were 5′-CGCCTCCGCG GTGGACGTAA AAAACGTCGT CAGCGTCGTC GTCGCGATGC AGAA TTCCGA-3′ and 5′-TCGGAATTCTGCATCGCGACGACGACGCTG ACGACGTTTT TTACGTCCACC GCGGAGGCG-3′, respectively. TAT-r-Aβ42 (r-tAβ42), r-Aβ42 containing TAT in its N-terminus, was produced using pET28b-GroES-ubiquitin-TAT-Aβ(42-1) which was constructed by using the method for pET28-GroES-ubiquitin-TAT-Aβ42. The sense and antisense primers for the PCR were 5′-TTGCGCCTCC GCGGTGGACG TAAAAAACGT CGTCAGCGTC GTCGTCGTCGCGC GATAGTCGTTGGTGGC-3′ 5′-GCCACCAACG ACTATCGCGC GACGACGACG CTGACGACGT TTTTTACGTC CACCGCGGAG GCGCAA-3′, respectively. Aβ42, r-Aβ42, tAβ42, and r-tAβ42 were purified from GroES-ubiquitin-peptide fusion proteins, as previously described [22]. Briefly, the fusion proteins produced in Escherichia coli were recovered in inclusion bodies that were solubilized in solubilization buffer (50 mM Tris–Cl, pH 8.0, 150 mM NaCl, 1 mM dithiothreitol [DTT], and 6 M urea). After the removal of insoluble proteins by centrifugation at 36,000× *g* and 4 °C for 30 min, the supernatant was diluted 2-fold with a buffer containing 50 mM Tris–Cl, pH 8.0, 150 mM NaCl, and 1 mM DTT. To remove GroES-ubiquitin, the fusion protein was digested with the Usp2-cc enzyme, as described in a previous study [23], followed by the addition of 100% methanol at a ratio of 1:1 to the sample. The desired peptide was recovered from the supernatant after centrifugation at 2000× *g* and 4 °C for 10 min. The peptide was monomerized by dissolution in 1,1,1,3,3,3,-hexafluoro-2-propanol (HFIP). The peptide obtained after the evaporation of HFIP was stored at −20 °C. We confirmed that the aggregates of the peptide (data not shown) were absent in the preparations in the previous and current studies [17,24]. Before use, the Aβ42 and tAβ42 peptides were dissolved at a concentration of 2 mg/mL in 0.1% NH_4_OH and 0.1% HCl, respectively, and the solution was sonicated for 10 min. The solution was then diluted to the desired concentration in phosphate-buffered saline (PBS) or cell culture media. The mass of the purified peptides was confirmed by a commercial peptide company (Anygen Co., Seongnam, Korea). Oligomers and fibrils of Aβ42 were prepared by incubating the peptides at 4 °C for 24 h and 37 °C for 24 h, respectively, as previously described [17,24].

### 2.3. Cell Culture and Cytotoxicity Assay

Human epithelial HeLa cells and human neuroblastoma SH-SY5Y were cultured as previously described [25]. For the cytotoxicity assay, the cells were seeded at a density of 15,000 cells/well in 96-well plates (Nunc, Roskilde, Denmark), cultured for 24 h, serum-deprived for an additional 12 h, and treated according to the treatment plan. Cell viability was assessed using the 3-(4,5-dimethylthiazol-2-yl)-2,5-diphenyltetarzolium bromide (MTT) reduction test [26], wherein 20 μL of a 5 mg/mL MTT solution in PBS was added to each well. After 2 h of incubation, 100 μL of solubilization buffer [20% sodium dodecyl sulfate (SDS) solution in 50% (*v*/*v*) N, N-dimethylformamide (DMF) (pH 4.7)] was added and the mixture was incubated for 12–16 h. Absorbance was recorded at 570 nm using a microplate reader (KisanBio, Seoul, Korea). Cytotoxicity was also determined using the alamarBlue assay [27,28], in which 10 μL of alamarBlue (Life Technologies, Inc., Carlsbad, CA, USA) was added directly to each well and the mixture was incubated for 4–16 h. Fluorescence was measured at excitation and emission wavelengths of 560 and 590 nm, respectively, using a Gemini-XS microplate spectrofluorometer (Molecular Devices, San Jose, CA, USA).

### 2.4. Immunocytochemistry

Cells (1 × 10^5^) were seeded in a 12-well plate, incubated for 24 h, and incubated for an additional 12 h in a serum-free medium at 37 °C. The cells were then treated with each of the Aβ42 peptides for the indicated times. The treated cells were fixed in methanol at −20 °C and permeabilized with 0.3% Triton X-100. After overnight blocking with 0.1% bovine serum albumin, mouse monoclonal anti-Aβ antibody 6E10 (BioLegend, San Diego, CA, USA) or rabbit polyclonal anti-caspase-9 (p10) antibody (Santa Cruz Biotechnology, Santa Cruz, CA, USA) was added to each sample and incubated overnight at 4 °C. After washing with PBS, Alexa-Fluor-546-TRITC-conjugated goat anti-rabbit IgG and Alexa-Fluor-488-FITC-conjugated goat anti-mouse IgG antibodies (dilution, 1:200, Invitrogen, Waltham, MA, USA) were added, followed by incubation for 2 h at room temperature and subsequent washing with PBS. Nuclei were stained with DAPI in Vectashield mounting medium (Vector Laboratories, Burlingame, CA, USA). Confocal images were obtained with a Carl Zeiss LSM510 microscope (Germany) using the manufacturer’s software (LSM 510), as previously described [29]. Four individual variable pinholes (97 µM) of 1.0 airy units for each confocal channel were used where a resolution was 2048 × 2048 pixels. The cells were focused using a plan apochromat 63 × 1.4 oil immersion objective and cells were projected at a single plane. Immunocytochemical analysis was conducted to measure the level of cells conveying Aβ peptide by counting the number of cells that had Aβ peptide accumulated intracellularly. In the image, 10 sectors were selected, were 3–6 cells were present per sector. The sectors were not selected if countable cells were not present. We did not consider zones with highly agglomerated cells to avoid counting errors.

### 2.5. Measurement of Caspase Activity

Cells (2 × 10^4^) were seeded in a 96-well plate, incubated at 37 °C for 24 h, and serum-starved for an additional 12 h. After treatment with the indicated Aβ peptides, the cells were washed twice with ice-cold PBS. Subsequently, 40 μL of lysis buffer (20 mM HEPES-NaOH, pH 7.0, 1 mM EDTA, 1 mM EGTA, 20 mM NaCl, 0.25% Triton X-100, 1 mM DTT, 1 mM PMSF, 10 μg/mL leupeptin, 5 μg/mL pepstatin A, 2 μg/mL aprotinin, and 25 μg/mL N-acetyl-Leu-Leu-Norleucinal) was added to each well. The mixture was incubated on ice for 20 min. Caspase assay buffer (20 mM HEPES-NaOH, pH 7.0, 20 mM NaCl, 1.5 mM MgCl2, 1 mM EDTA, 1 mM EGTA and 10 mM DTT) and Ac-DEVD-amino-methyl-coumarin (AMC) (AG Scientific, Inc., San Diego, CA, USA) were then added to the mixture. The release of AMC was monitored for 1 h at 5 min intervals at excitation and emission wavelengths of 360 and 480 nm, respectively, using a Gemini-XS microplate spectrofluorometer (Molecular Devices, San Jose, CA, USA), as previously described [30].

### 2.6. Western Blot Analysis

Cells (4 × 10^5^) were cultured in a 60 × 15 mm culture dish and incubated for 24 h at 37 °C; then, they were serum-deprived for another 12 h before treatment. After treatment according to the plan, the cells were harvested, washed with ice-cold PBS, and resuspended in lysis buffer (50 mM Tris–HCl, pH 8.0, 150 mM NaCl, 1% Triton X-100, 5 mM EDTA, 5 mM EGTA, 1 mM PMSF, 10 μg/mL leupeptin, 2 μg/mL pepstatin A, and 2 μg/mL aprotinin). After a 20 min incubation on ice, the supernatant was obtained from the lysed cells after microfuge centrifugation at 18,000× *g* at 4 °C for 15 min. Equal amounts of proteins (measured using the Bradford assay) were subjected to 12–15% SDS-polyacrylamide gel electrophoresis (PAGE) as described in an earlier report [31] and then the proteins were transferred to a polyvinylidene fluoride membrane. The membrane was immunoprobed with mouse monoclonal anti-lamin A/C, lamin B1, and β-actin (Santa Cruz, CA, USA) and then with horseradish peroxidase-conjugated secondary antibodies (Santa Cruz Biotechnology, Santa Cruz, CA, USA) [29]. The blots were visualized using WESTER ηC ULTRA (Cyanagen, Bologna, Italy).

### 2.7. Fibrillogenesis

The peptide solution (20 µM, 300 μL) prepared in PBS was incubated at 37 °C. At each time point, the incubated peptide solution was mixed properly by pipetting and then 20 μL solution was taken and mixed with 80 μL of freshly prepared 5 μM thioflavin T (ThT, Bioneer, Daejeon, Korea) in PBS. The resulting fluorescence was measured on a microplate spectrofluorometer Gemini-XS (Molecular Devices, San Jose, CA, USA) at an excitation wavelength of 445 nm and emission wavelength of 490 nm, as described earlier [9].

### 2.8. Analysis of Secondary Structure

The peptide solution was prepared in PBS (20 μL), and the spectra were recorded immediately after incubation at 37 °C. Far UV circular dichroism (CD) spectra were recorded with a 1 mm path length cuvette at 0.5 nm intervals between 190 and 250 nm using a Jasco J-810 Spectropolarimeter (Jasco Co., Gunma, Japan) at 25 °C. Five accumulative readings were acquired at a 0.1 nm resolution, 0.5 s response time, and 50 nm/min scan speed [19].

### 2.9. Transmission Electron Microscopy (TEM)

A 5 μL sample was loaded on a Formvar-coated 200-mesh nickel grid (SPI Supplies, West Chester, PA, USA) and kept for 5 min. The extra solution was drained from the grid and washed thrice with distilled water. The sample was then negatively stained with 2% uranyl acetate for 1 min. The grids were focused under a TEM (H-7600, Hitachi, Tokyo, Japan) operated at an accelerating voltage of 80 kV and a magnification of 40,000× [24,32].

## 3. Results

### 3.1. Cytotoxicity of Wild-Type Aβ42 and tAβ42

In the current study, we initially explored the cytotoxic effect of each Aβ42 preparation (see below) and then determined its correlation with the intracellular location of the selected preparations to explore the relationship between the two factors. We chose purified Aβ42 peptides over other options, such as the intracellular expression of the peptide wherein quantitative and comparative studies on intracellular and extracellular localization of peptides will be difficult. Freshly prepared Aβ42 (monomeric preparation, mAβ42) and an oligomeric preparation of Aβ42 (oAβ42) were used in this study. We did not test insoluble fibrillar aggregates of Aβ peptides in the current study because the relatively less cytotoxic species [17,24] were not thought to be cell-permeable. To facilitate the study, we also constructed and tested cell-permeable Aβ synthesized by attaching the cell-permeable peptide TAT to the N-terminus of the Aβ42 peptide (tAβ42). Herein, we used different cells, including human neuroblastoma SH-SY5Y and epithelial HeLa cells. Most of the results with both cells were comparable (Appendix A), except that the SH-SY5Y cells showed low apoptotic caspase activation and were easily killed compared with HeLa cells. This resulted in difficulties in the analyses necessary for this study. Thus, the results reported here were mostly from HeLa cells.

Cell viability was assessed by the MTT assay, and results were confirmed using the alamarBlue assay because MTT formazan production can be decreased by Aβ treatment without overt cell death [33]. mAβ42 at up to 20 µM induced less than 10% cell death after 12 h of incubation (open circles with a solid line in Figure 1A,B, the symbols of the figures were partially overlapped with closed circles). Further incubation (24 h) of the cells with the peptide increased cell death by up to 40% (open circles with a solid line in Figure 1C,D). oAβ42 was not cytotoxic after 12 h of incubation (closed circles with a solid line in Figure 1A,B), but it showed cytotoxicity when the cells were further incubated for 24 h (closed circles with a solid line in Figure 1C,D). Consistent with previous reports [34], oAβ42 was more cytotoxic than mAβ42 (Figure 1C,D); for instance, ~75% cell viability was observed with 5 µM oAβ42, whereas >10 µM concentration of mAβ42 was necessary for a similar level of cytotoxicity (Figure 1C,D). Considering a report indicating that oligomerization of the peptide increases cytotoxicity by up to 10 folds [34], the difference was less than expected. We speculate that conformational transformation of mAβ42, during incubation, to other structures could increase the cytotoxicity of the preparations.

The double treatment assays were also employed here because we wanted to test diverse cell treatment conditions for the current study. The double treatment of cells with peptides increases cell death [29,35] and apoptotic caspase activation and Aβ-specific lamin fragmentation were observed only in cells subjected to double treatment [29]. The underlying mechanisms are not clearly understood, although it is speculated that the nucleation process of Aβ42 polymerization is necessary for death signal transduction [29,35]. In the assay, cells were first incubated with the indicated concentrations of Aβ42 for 2 h, washed with the culture media to remove the added peptide, and then treated with a new preparation of the same concentration of Aβ42 for 10 h (2 + 10 h sample) or 22 h (2 + 22 h sample). No prominent increase in cytotoxicity was detected in double treatment equivalent to 12 and 24 h (2 + 10 h and 2 + 22 h) samples with mAβ42 (open circles with a dotted line in Figure 1A–D, the symbols of the figures were partly overlapped with those of 12 and 24 h mAβ42 samples). However, a significant increase in cytotoxicity was observed in samples treated twice (2 + 10 h) with oAβ42 (Figure 1A,B), and the level was further enhanced in samples incubated for (2 + 22) h (Figure 1C,D). It seemed that Aβ42 species with robust cytotoxicity were quickly formed in the 2 + 10 h samples with oAβ42, whereas their formation might have taken a longer time in other singly-treated samples. It is also possible that a cytotoxicity-enhancing process, such as polymerization of the peptide [36,37], occurred more quickly in the doubly-treated samples than in the other singly-treated samples. The identity of the ‘super-toxic’ species or process has not been elucidated, although the oligomeric form [6,7,8] and/or the polymerization process [36,37] have been suggested as underlying factors.

tAβ42 was more cytotoxic than the wild-type Aβ42 preparations in the 12 and 24 h-incubation samples (Figure 1A–D). Treatment of cells with freshly prepared tAβ42 (mtAβ42) at 2.5~5 µM for 24 h resulted in ~50% viability in both assays (open triangles with a solid line in Figure 1C–F), whereas >20 µM concentration of wild-type oAβ42 was necessary to induce these levels of viability (Figure 1C–F). Interestingly, oligomeric preparations of the fusion peptide [otAβ42, solid line without symbols in Figure 1A–D; overlapped with those of mtAβ42] or the double treatment (broken line for mtAβ42 and dotted lines for oAβ42 in Figure 1A–D are overlapped with those of mtAβ42) did not increase the cytotoxicity of the peptide preparations. This implies that tAβ42 has a structure that wild-type Aβ42 gained by oligomeric preparation and double treatment. As the cytotoxicity of oAβ42 and mtAβ42 did not increase with prolonged incubation up to 48 h (Figure 1E,F), the cells were treated with the peptides for less than 48 h in the following experiments. As controls, the cytotoxicities of r-Aβ42 and r-tAβ42 were also explored. Cells incubated with either peptide at 20 µM for 24 h showed >90% cell viability and the cytotoxicity difference between the two peptides was less than 3% (Figure 1C,D), indicating that the strong toxicity of tAβ42 is not due to the toxicity of TAT sequence itself. Hereafter, tAβ42 indicates mtAβ42, which was used in most of the subsequent studies, unless otherwise indicated.

### 3.2. Internalization of Aβ42 into the Cells

Next, we counted cells conveying Aβ peptide intracellularly after treatment with each peptide preparation, using a confocal microscope. If cells intracellularly accumulate the peptide that was added extracellularly, it means that it means that the peptide entered the cells, possibly indicating the cell permeability of the peptide [15]. For samples treated for 12 h with 20 μM mAβ42 and oAβ42, ~5% of cells contained the peptide (see Figure 2B,E for representative images showing cells without the peptides inside and Figure 2I for summary). Longer incubation of the cells for 24 h with the peptide resulted in more cells conveying the peptide (~17% for mAβ42 and ~26% for oAβ42) (Figure 2C,F,I). The samples treated twice with oAβ42 for 2 h and subsequently for 10 h, which showed high levels of cell death (Figure 1), showed higher levels of internalization of the peptide (~23%, Figure 2G) than those treated with oAβ42 for 12 h (Figure 2E) and 2 + 10 h samples treated with mAβ42 (Figure 2D). This implied that the double treatment with the oligomeric preparations increased the permeability of the peptide (see Figure 2I for summary). On the other hand, >95% of cells treated with 5 µM tAβ42 for 12 h, which showed a similar level of cell death as those treated with 20 µM oAβ42, intracellularly contained the peptide (Figure 2H,I). It was expected that more cells became dead with the high level of intracellular location of tAβ42. We speculate that a certain level of intracellular accumulation of the peptide is necessary to induce cell death because the higher concentrations of the peptide showed stronger cytotoxicity (Figure 1) even with a similar level of intracellular peptide (data not shown). Samples incubated for longer times, such as 48 h and 2 + 22 h, are not shown here, because the images of those samples were not clear, probably due to cell debris. In summary, the intracellular levels of Aβ peptides in the cells roughly correlated with the cytotoxicity of the different preparations (Figure 2I). It is noteworthy that tAβ42 appeared to interact with caspase-9, similar to wild-type Aβ [26], as shown in the confocal assay (see the yellowish color of the overlapping image in Figure 1H).

### 3.3. Effect of the Inhibition of Aβ42 Cytotoxicity on Cell Viability and the Internalization of Aβ42 into Cells

The relationship between the intracellular location and cytotoxic features of Aβ peptides was further explored using known inhibitors of Aβ cytotoxicity. Two of the several inhibitors of Aβ cytotoxicity tested, taiwaniaflavone and MBP, which we extensively characterized previously [17,18], were explored. We chose the inhibitors because they are effective inhibitors of cytotoxicity and their inhibitory mechanisms are different (see below). Taiwaniaflavone, at ~10 µM concentration, increased the viability of cells treated with oAβ42 by ~1.4 fold (Figure 3A,B), which was consistent with previous results [17]. The viabilities of cells treated with tAβ42 and doubly with oAβ42 were also enhanced by up to ~1.6 fold (Figure 3A,B). MBP had a protective effect on cells treated with oAβ42 singly or doubly, increasing the cell viability by ~1.4 fold; however, it was only when oAβ42 was pre-incubated with the protein (Figure 3C,D). Without pre-incubation, cell viability increased by less than ~1.2 fold (Figure 3C,D). However, for tAβ42, MBP showed little protective effect (<20%) even in the pre-incubated samples (Figure 3C,D). We focused on investigating the relationship between the cytotoxicity and intracellular location of Aβ peptides in the following study, rather than exploring the underlying mechanism for the differential effect of MBP on cells treated with the peptides. 

The effects of taiwaniaflavone and MBP on the internalization of oAβ42 and tAβ42 into the cells were determined by measuring the intracellular levels of peptides using a confocal microscope, as shown in Figure 2. Double treatment with oAβ42 was not used in the assay because the sample was highly viscous with the inhibitor, which disturbed the clarity of the confocal images. Approximately 14% and 16% of cells treated with oAβ42 and tAβ42 for 24 h in the presence of taiwaniaflavone conveyed the intracellular peptide, respectively (Figure 4A,D show cells without the peptide intracellularly, and Figure 4F is a summary of results). This level was low, compared with ~26% and >95% of that obtained with the same treatment without the inhibitor (Figure 2F,H, and Figure 4F). With taiwaniaflavone, the viability of the cells treated with oAβ42 and tAβ42 increased by ~1.4- and ~1.6-fold, respectively (Figure 3A,B), and cell death reduced to ~17% and ~19% from ~39% and ~44% observed in the cells treated without the inhibitor (Figure 4F). This confirmed the correlation between cell death and the internalization of the peptide into the cells.

MBP caused a reduction in the internalization of oAβ42 (~16% cells [Figure 4C] vs. ~26% in samples without the protein (Figure 2F), summarized in Figure 4F]; however, this occurred only when the protein was pre-incubated with the Aβ42 peptide. The MBP protein barely reduced the internalization of tAβ42 (>85% (Figure 4E,F) vs. >95% (Figure 4F) in samples without the protein), even though the peptide was pre-incubated with the protein. The inhibitory patterns of the internalization of each peptide by MBP were consistent with those of the cell death results shown in Figure 3 (summarized in Figure 4F), indicating again a strong relationship between cell death and the intracellular location of Aβ peptides. Taiwaniaflavone inhibits Aβ42 fibrillogenesis to accumulate nontoxic off-pathway Aβ oligomers [17], while MBP decreases the active concentration of Aβ42 by sequestering it as Aβ42-MBP complex to suppress ongoing nucleation [18]. Although their inhibitory mechanisms of the cytotoxicity are different, the consequences were decreasing the intracellular location of wild-type Aβ42 with the suppression of cytotoxicity. This is also true for taiwaniaflavone with tAβ42. Currently, the molecular mechanism for the inhibitory effect of the above inhibitors is not known. One hypothetical explanation is that the inhibitors potentially block the binding of Aβ peptides to the cell membranes as shown before [38].

### 3.4. Lamin Protein Fragmentation and Caspase Activation Induced by tAβ42

We determined whether the cell-permeable tAβ42 is able to induce the same cytotoxic process as the wild-type peptide. Otherwise, the highly toxic property of the peptide may be due to other cytotoxic pathways than the intracellular location. Previously, we showed that the oligomerization process and double treatment, which are necessary for the high level of cytotoxicity and internalization of wild-type Aβ42 into the cells (Figure 1 and Figure 2), are essential for inducing apoptotic caspase activation, a hallmark process of caspase-dependent apoptosis [39,40,41], and Aβ-specific lamin protein fragmentation [29]. Thus, to examine whether tAβ42 can induce the same cytotoxic process as the wild-type Aβ peptide, caspase-3/7-like DEVDase activation and lamin fragmentation were explored in the fusion peptide-treated cells. 

The mtAβ42 and otAβ42 preparations induced DEVDase activity in a dose-dependent manner, with the highest activation observed after 36 h of incubation (Figure 5A–C). Oligomerization and double treatment were confirmed to be necessary for the wild-type peptide to induce a robust activation of the enzyme (Figure 5A), consistent with previous reports [29,35], while mtAβ42 or otAβ42 induced the activity similarly independently of the double treatment. These results are compatible with those of the MTT and alamarBlue cytotoxic assays shown in Figure 1. 

The effects of taiwaniaflavone and MBP on DEVDase were monitored in cells treated with oAβ42 and tAβ42 for 2 + 22 h and 36 h, respectively, during which time the activity was prominent (Figure 5B,C). In the presence of taiwaniaflavone, DEVDase activity was reduced in both the cell samples (Figure 5D). The differential effects of MBP on cell death and the internalization of Aβ peptide (Figure 3 and Figure 4) were also observed from the results of the DEVDase assay. The activation-induced by oAβ42 treatment for 2 + 22 h reduced when MBP was pre-incubated with the peptide, but such inhibition was not observed with tAβ42 (Figure 5D).

Investigation of lamin protein fragmentations is a useful tool for the exploration of Aβ-specific cytotoxic processes, because the Aβ peptide induces specific lamin fragmentations, resulting in the generation of ~46 kDa N-terminal and ~21 kDa C-terminal fragments from lamin A and B, respectively. It has been reported that the fragmentations occur only in cells treated twice with oAβ42 [29]. Consistently, in this study, these fragments were generated in cells treated with 20 µM oAβ42 for 2 + 22 h (Figure 5E). They were not detected in the cells treated in the same way in the presence of taiwaniaflavone and with oAβ42 pre-incubated with MBP (Figure 5E), indicating that these inhibitors suppressed the process leading to the fragmentations. Again, pre-incubation is necessary for the inhibitory effect of MBP (compare lanes M and pM in Figure 5E).

The lamin protein fragmentations occurred earlier than caspase activation in the Aβ42-induced cytotoxic process [29]. Thus, for tAβ42, it was examined after 24 h of incubation instead of 36 h of the enzyme assay. The peptide also led to the specific fragmentations of lamin A and B in the cells (Figure 5F). The patterns of the fragmentations were the same as those of wild-type Aβ42 shown in Figure 5E. The fragments were not detected in cells treated with tAβ42 in the presence of taiwaniaflavone (Figure 5F), which was similar to the results for oAβ42. The effect of MBP on tAβ42-induced fragmentation was different from that of wild-type Aβ42 in that pre-incubation of tAβ42 with MBP led to the generation of lamin fragments (see lanes M and pM of Figure 5F), which was consistent with the results of the analyses of cytotoxicity (Figure 3C,D), internalization (Figure 4E), and DEVDase activity (Figure 5D).

The patterns of tAβ42-induced fragmentation of lamin proteins were only observed in cells treated with oAβ42 [29]. Thus, we speculated that the fusion peptide has biological properties similar to those of the wild-type peptide conformer. The results of the inhibitor studies on oAβ42- and tAβ42-induced processes, such as caspase activation (Figure 5D), lamin fragmentation (Figure 5E,F), and cytotoxicity (Figure 3 and Figure 4F) were consistent with those of internalization of the peptides into the cells (Figure 4A–E), confirming the close relationship between the cytotoxicity and intracellular location of the peptides.

### 3.5. Structural Characterization of tAB42

Polymerization of Aβ42 is necessary for the cytotoxicity of the peptide [36,37]. In the polymerization process, conformational changes in β-sheets occurred, followed by oligomerization and fibrillar formation. We determined whether this process would also occur for the tAβ42 peptide because these experiments could provide biophysical clues for understanding the association between polymerization and cytotoxicity in wild-type Aβ42. We initially examined the fibrillogenic kinetics of tAβ42, which was compared with that of wild-type Aβ42. In the assay, tAβ42 was compared with wild-type Aβ42 at a 20 µM concentration, because aggregation kinetics of the latter peptide was easily measured at the concentration [19]. As the polymerization kinetics of Aβ peptide is not dependent on concentration [17,42], data obtained for 20 µM concentration of the peptide could apply to other concentrations to a certain extent. tAβ42 did not show saturation kinetics even after 48 h of incubation, in the ThT binding assay (Figure 6A). This observation was expected because fibrils of tAβ42 were not detected after incubation for up to 48 h (data not shown).

ThT cannot detect Aβ intermediates [9,43]; thus, to improve our understanding of the structural transformation of tAβ42, we determined the secondary structure of the peptide using CD. Again, a 20 µM concentration of the peptide was used to obtain a strong signal, which was compared with that of the wild-type peptide. Freshly prepared tAβ42 exhibited negative ellipticity at ~195 nm (Figure 6C), which was comparable to that of mAβ42 (Figure 6B). After incubation at 37 °C, the wild-type peptide reached a maximal negative ellipticity at ~217 nm after 12 h, which was maintained at 24 h (only 24 h data are shown in Figure 6B) [17]. Similarly, tAβ42 showed negative ellipticity at ~217 nm and 24 h. However, the peptide did not exhibit any changes at 12 h (broken line of Figure 6C, overlapped with 0 the line) and the level of reduction was less than that of the wild-type peptide (Figure 6C). These changes arise from the formation of β-sheet structures in peptides [22]. Thus, this result implies that the transition of the random coil to the β-sheet structure is slower, and the level is lower in tAβ42 peptide samples than in the wild-type peptide samples.

We further characterized the conformational transition of tAβ42 using TEM to directly examine the formed structural species. However, only a rough estimation of the level of the formed aggregates was possible in the TEM analysis. Thus, the numbers below are approximated ones and were presented just to examine if the results are consistent with those of fibrillogenesis (Figure 6A) and those of β-sheet formation (Figure 6B). Furthermore, only parts of the TEM images could be presented in the figures, because the entire images were too big. The images in Figure 6D,E were selected from the whole images to show clearly the conformational species and to compare the shape of aggregates formed from wild-type and tAβ42 peptides. The result with a 5 µM concentration of tAβ42 is shown because the conformational species formed at the concentration of the peptide were seen well by TEM and adopted for the above experiments (Figure 2, Figure 4 and Figure 5). That with 20 µM tAβ42 was similar in the level (after normalization) and types of the formed conformation species. It appears that a certain amount of the wild-type Aβ42 was transformed to oligomeric or protofibrillar structures upon incubation at 4 °C for 24 h (upper and middle panels of Figure 6D, showing only selected parts of images of oligomer-like and protofibrils-like structures), which was consistent with the results of previous studies [44,45,46]. Similar conformational species were also detected when tAβ42 was incubated under the same conditions (upper and middle panels of Figure 6E, showing only selected parts of images of oligomer-like and protofibrils-like structures). However, the levels of formed species were much lower (roughly ~1/5) than those of the wild-type at the same concentration (data not shown), although the exact determination of the levels of the formed structure was difficult. Wild-type Aβ42 aggregated to form fibrillar species upon incubation at 37 °C for 24 h, as expected (lower panel of Figure 6D). Similar species were also observed in the tAβ42 sample incubated under the same conditions (lower panel of Figure 6E). The level of these aggregates of tAβ42 was <10% of the wild type at the same concentration (data not shown), which was consistent with the results shown in Figure 6A. On SDS-PAGE typical SDS-resistant dimeric or trimeric species observed in wild-type Aβ42 preparations [17] were not detected in tAβ42 preparations, which were mostly monomeric on the gel (data not shown). Altogether, these observations indicated that the conformational transition (to β-sheet) and fibrillogenesis of tAβ42 were partial and slower than those of the wild-type peptide, and this property may contribute to the solubility of tAβ42.

## 4. Discussion

The level of Aβ42 peptide internalization into the cells (Figure 2) was high for peptide preparations with strong cytotoxicity (Figure 2IFigure 4F), Aβ-specific lamin protein fragmentation (Figure 5), and a high level of caspase activation (Figure 5). As the peptide was administered extracellularly in this study, unlike in the previous study where the peptide was expressed intracellularly [14], the internalization possibly indicated the cell permeability of the peptide [15]. The following studies showed that the highly cell-permeable tAβ42 had superior cytotoxicity (Figure 1 and Figure 2), and inhibitors that suppressed Aβ42- and tAβ42-induced cell death, lamin protein fragmentation, and caspase activation (Figure 3, Figure 4 and Figure 5) reduced the level of internalization. These results support the close relationship between the cytotoxicity and cell permeability of the peptides. The inhibitors of lamin fragmentation [29] and caspase activation did not effectively hinder the internalization of the peptide into the cells (data not shown); hence, the intracellular location of the peptide appears to be an upstream event leading to Aβ-specific lamin fragmentation and caspase activation. Thus, we speculate that Aβ-specific lamin fragmentation and caspase activation are induced by Aβ peptides located intracellularly.

The Aβ42 peptides might enter the cell after the cell toxicity developed first. However, we think that the intracellular location of the peptides resulted in cell death. One speculation supporting the assertion is that the cytotoxicity-enhancing effect of the TAT sequence attached to Aβ42 should not be due to the toxicity of the sequence itself, but due to its sequence’s ability to promote locating the fused tAβ42 peptide intracellularly. This is because r-tAβ42 showed negligible cytotoxicity and was only <3% more toxic than r-Aβ42 at the same concentration (Figure 1C,D). Wild-type Aβ42 shares the cell death signal transduction pathway with the TAT fusion peptide, as both peptides showed Aβ-specific lamin fragmentation (Figure 5E,F). Thus, the above suggestion could be also applicable to the wild-type peptide.

It is reasonable that the overall cytotoxicity of Aβ is possibly the ‘combined sum’ of the toxicity of all the Aβ conformational species, such as monomers, oligomers, and fibrils [17,24]. It is also possible that super-toxic Aβ species play a major role in the peptide-induced cytotoxic processes. However, the super-killer model and the ‘combined sum model’ are not exclusive, because the super-killer species may be the strongest contributor to cytotoxicity among all the cytotoxic species. Further study is essential to determine which model is more applicable to the pathogenesis of AD. The results of the current study are compatible with the ‘super-killer model’ because highly cytotoxic Aβ species or processes were identified. The followings are discussed based on the model.

tAβ42 induced Aβ-specific lamin protein fragmentation, indicating that the peptide shares the cytotoxic signal transduction pathway with the wild-type peptide (Figure 5F), and caspase activation with a single treatment and without oligomer enrichment (Figure 1 and Figure 5). Furthermore, tAβ42 was highly soluble and barely formed fibrils (Figure 6), which could contribute to minimal cytotoxicity loss due to the completion of fibrillogenesis (Figure 6) [17,24]. These results imply that tAβ42, without any further preparative processes, has the properties that wild-type Aβ42 gains through oligomer enrichment and double treatment. Thus, we speculate that tAβ42 has the properties of the hypothetical Aβ super killer of the above model, some of which would be high cell permeability and solubility. tAβ42 should be useful to screen agents that can suppress Aβ cytotoxicity and explore the underlying mechanisms associated with the role of Aβ cytotoxicity in AD pathogenesis because it does not need the complex preparation processes such as oligomerization and double treatment.

Based on the findings of the current study showing the close relationship between the cytotoxicity and intracellular location of the Aβ peptide, we suggest that blocking the internalization or reducing the cell permeability of this peptide could be an efficient way to reduce Aβ cytotoxicity (Figure 3 and Figure 4). The results with two inhibitors tested in the current study should be informative in this context, although the mechanistic details regarding these observations were not explored further except in the study of the relationship between cytotoxicity and intracellular location of Aβ. Re-evaluation of other inhibitors of Aβ cytotoxicity in terms of the intracellular location of the peptide will be interesting. The lines of study with the data presented here will give new information regarding the cytotoxicity of the peptide as well as develop a new strategy for AD therapeutics.

## Figures and Tables

**Figure 1 life-12-00577-f001:**
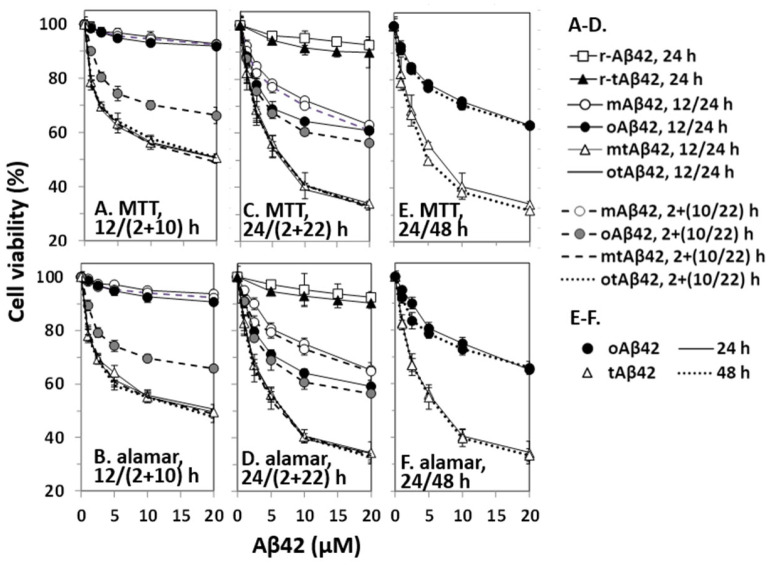
Cytotoxicity of Aβ42 and tAβ42. HeLa cells were treated with mAβ42, oAβ42, mtAβ42, and otAβ42 at the indicated concentrations for 12/(2 + 10) h (**A**,**B**), 24/(2 + 22) h (**C**,**D**) and 24/48 h (**E**,**F**). After treatment, cell viability was assessed with the MTT reduction assay and alamarBlue assay. Results are expressed as the mean ± standard deviation of values from three independent experiments. Data for Aβ42 12 h, mAβ42 2 + 10 h and oAβ42 12 h in A and B; mtAβ42 12 h, mtAβ42 2 + 10 h, otAβ42 12 h and otAβ42 2 + 10 h in A and B; mtAβ42 24 h, mtAβ42 2 + 22 h, otAβ42 24 h and otAβ42 2 + 22 h in C and D are overlapped.

**Figure 2 life-12-00577-f002:**
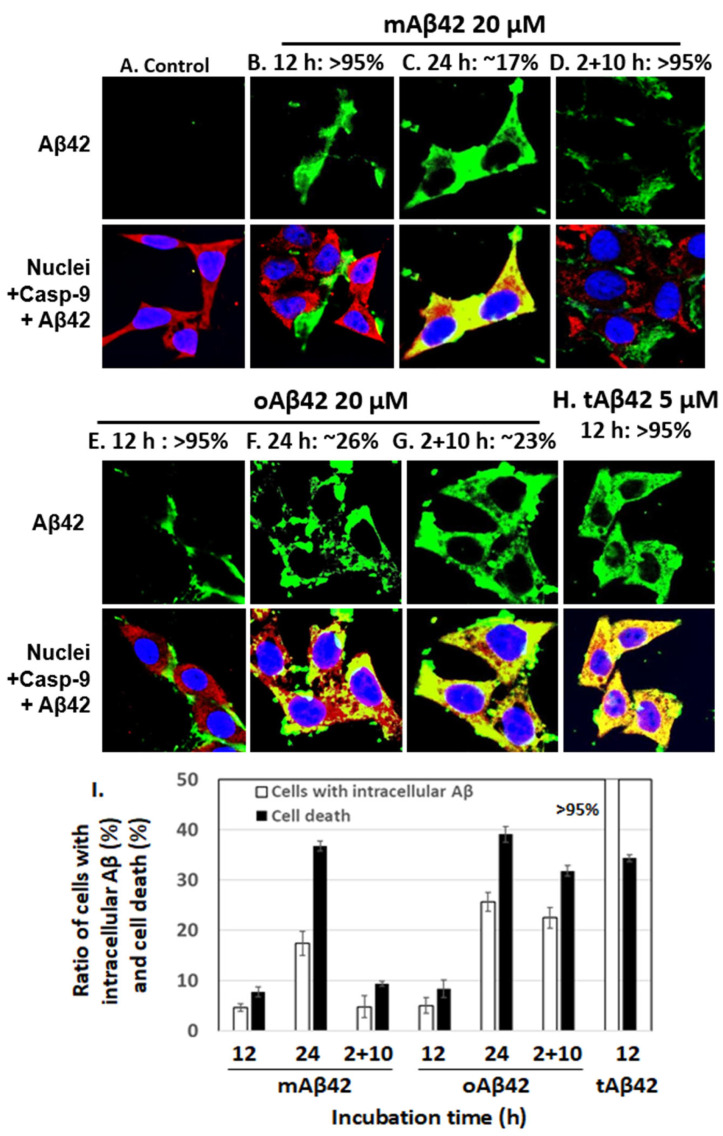
The extent of cellular internalization of Aβ42 and tAβ42. (**A**–**H**) HeLa cells were treated with Aβ42 species as indicated, except for the control. Next, the confocal microscopic images of the cells were taken for Aβ and caspase-9 by applying mouse anti-Aβ (6E10) and rabbit anti-caspase-9 (p10) antibodies. Aβ (green) and caspase-9 (red) were visualized using the secondary antibodies indicated in the Section 2. Nuclei were stained with DAPI (blue). Caspase-9, which was monitored to locate the cytoplasm, was previously shown to interact with the Aβ42 peptide [26]. Thus, yellow spots appear to be the results of the interaction of caspase-9 and Aβ. The images that we need to examine closely were presented here; the images for Aβ peptides located extracellularly are presented in (**B**,**D**,**E**), while those for intracellular peptides are shown in (**C**,**F**–**H**). The numbers on the upper side of the figures indicate the percentages of cells with Aβ peptide, extra- or intracellularly, according to the images. At least three independent experiments were carried out and only the representative images of cells are displayed. (**I**) Summary of the comparison of the number of cells with intracellular Aβ and cell death for both peptides is presented. The number of cells with intracellular Aβ was calculated using the results shown in Figure 2B–H, and the data for cell death were obtained from Figure 1. Data are presented as the mean ± standard deviation of values from three independent experiments.

**Figure 3 life-12-00577-f003:**
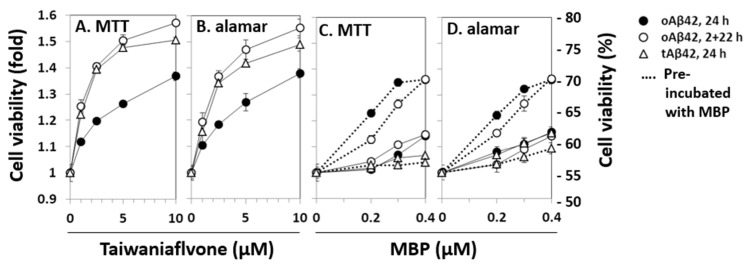
Effects of taiwaniaflavone and MBP on the cytotoxicities of Aβ42 and tAβ42. HeLa cells were treated with 20 µM oAβ42 and 5 µM tAβ42 in the absence and presence of the indicated concentrations of taiwaniaflavone (**A**,**B**) or MBP (**C**,**D**). In the pre-incubated samples, the indicated peptides were incubated with MBP at 37 °C for 12 h before the treatment (dotted lines of **C**,**D**) [18]. Cell viability was assessed using the MTT reduction assay and alamarBlue assay as described in Figure 1. Results are expressed as the mean ± standard deviation of values obtained from three independent experiments.

**Figure 4 life-12-00577-f004:**
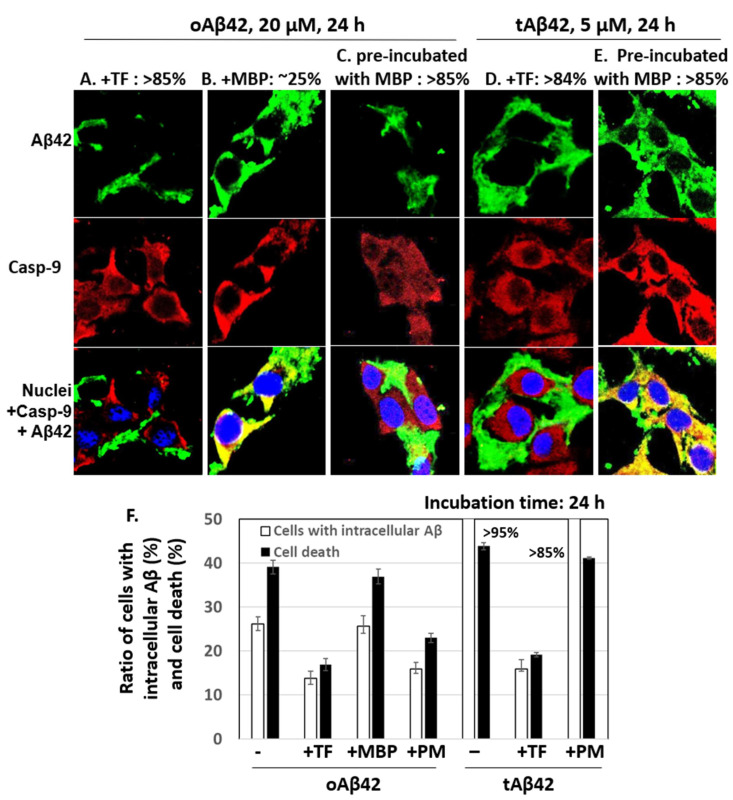
Effects of taiwaniaflavone and MBP on cellular internalization of Aβ42 and tAβ42. (**A**–**E**) HeLa cells were treated with 20 µM oAβ42 or 5 µM tAβ42 for 24 h in the presence of either 10 µM taiwaniaflavone or 0.4 µM MBP (Figure 3), and the confocal microscopic images of the cells were taken for Aβ and caspase-9 by applying mouse anti-Aβ (6E10) and rabbit anti-caspase-9 (p10) antibodies. Aβ (green) and caspase-9 (red) were visualized using the secondary antibodies indicated in the Section 2. The pre-incubation was performed as shown in Figure 3. Images for Aβ peptides located extracellularly are shown in (**A**,**C**,**D**), while those for intracellular peptides are in B and E. The numbers on the upper side of the figures indicate percentages of cells having Aβ peptide extra- or intracellularly as shown in the images. At least three independent experiments were carried out, and only representative images of the cells are displayed. (**F**) Summary of the comparison of the number of cells with intracellular Aβ and cell death. The number of cells with intracellular Aβ was calculated using the results from Figure 4 and data for cell death were from Figure 3. Data are presented as the mean ± standard deviation of values from three independent experiments. TF and PM stand for taiwaniaflavone and pre-incubation with MBP, respectively.

**Figure 5 life-12-00577-f005:**
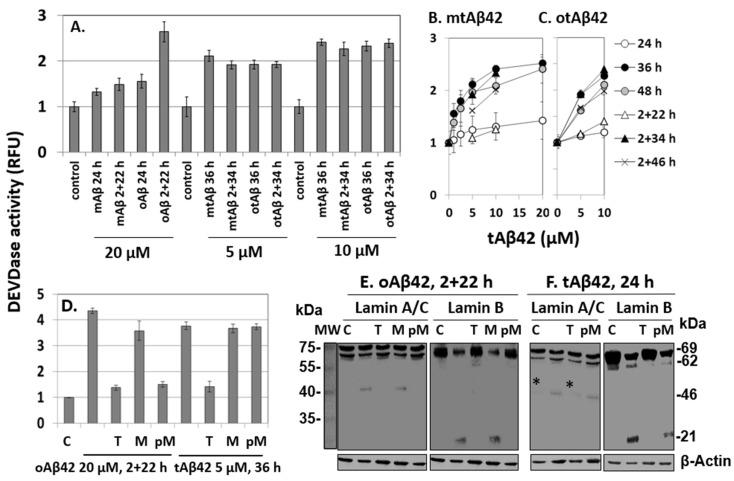
DEVDase activity and fragmentation of lamin A/C or B induced by Aβ42 and tAβ42. (**A**–**D**) HeLa cells were treated with the indicated Aβ42 peptides for the indicated period, and DEVDase activity was measured with 10 µM ac-DEVD-AMC substrate. RFU indicates a relative fluorescence unit. D, effects of 10 µM taiwaniaflavone and 0.4 µM MBP on DEVDase activity. Results are expressed as the mean ± standard deviation of values from three independent experiments. (**E**,**F**) HeLa cells were treated in the same condition as described in Figure 5D in the absence and presence of taiwaniaflavone or MBP. Next, cell lysates were prepared and the fragmentation of lamin A/C or lamin B was evaluated via western blotting. A faint band denoted by * in panel F was seen in the control sample as well; hence, it was concluded that it was not a fragment from lamin A/C. β-Actin was used as the loading control. Relative molecular weights are denoted at the right in kDa. MW is the molecular weight of marker. The result is representative of at least three independent experiments. Control or C is cells incubated without Aβ peptide. (**D**–**F**) M, PM, and T indicate 0.4 µM MBP, pre-incubation with 0.4 µM MBP, and 10 µM taiwaniaflavone, respectively, administered with the Aβ42 peptide to cells, as in Figure 3.

**Figure 6 life-12-00577-f006:**
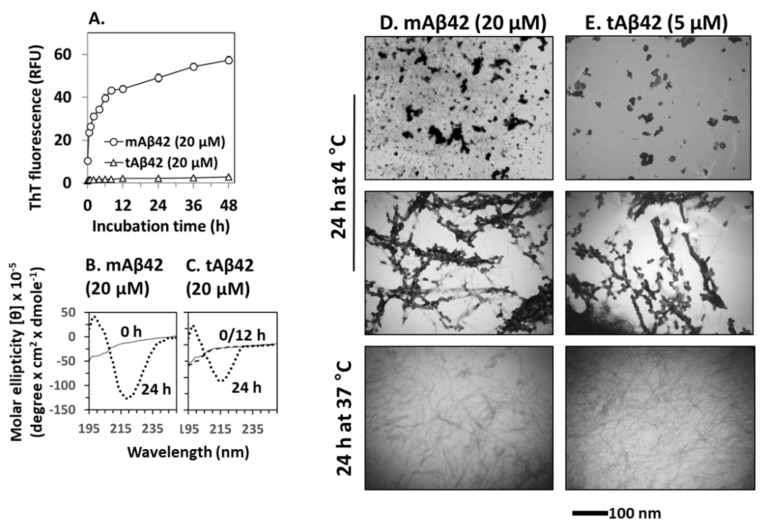
Physical characterization of tAβ42. (**A**) Fibrillogenesis of tAβ42 and mAβ42 was monitored by measuring ThT fluorescence of the peptide at the indicated incubation time at 37 °C. RFU stands for relative fluorescence unit. (**B**,**C**) CD spectra were recorded for freshly prepared sample (solid line), 12 h-incubated (broken line overlapped with that of freshly prepared sample; shown only for tAβ42), and 24 h (dotted line)-incubated samples of mAβ42 and tAβ42. (**D**,**E**) mAβ42 and tAβ42 peptides were incubated under the indicated conditions and images were captured using TEM at a magnification of 40,000× tAβ42 samples contained low levels of polymerized species (see the results). The parts of TEM images were presented to show the conformational species clearly. These are representative ones obtained from at least three independent experiments. Scale bars are shown beneath the TEM images.

## Data Availability

Not applicable.

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
