# Peer review of "Evidence for a Strong Relationship between the Cytotoxicity and Intracellular Location of β-Amyloid"

_life, 2022, doi:10.3390/life12040577_

Round 1

Reviewer 1 Report

Please consider the file attached.

Author Response

Major issues:

1) The title of the manuscript is quite vague and speculative. Indeed, the correlation between the cell permeability and cytotoxicity in Aβ-peptide amyloid species has been widely studied and thoroughly elucidated in several important studies that are completely ignored in this manuscript (Some examples are: J Mol Biol. 2009 Feb 13;386(1):81-96 doi: 10.1016/j.jmb.2008.11.060; Sci Rep 6, 32721 (2016) doi:10.1038/srep32721; Front Cell Neurosci. 2019 Jul 18;13: 309 doi: 10.3389/fncel.2019.00309). In this respect, this study does not add new information in the correlation between membrane permeability and cell toxicity in Aβ-amyloid species.

à We thought that for the intracellular location of Aβ peptide it should enter cells, which implies the permeability of the peptide. However, we measured the intracellular location of Aβ peptide in the study. Thus, the word “permeability” of the MS title, abstract, introduction, results and discussion was changed to “intracellular location” to indicate clearly what we did. The studies this reviewer gave were mostly about the interaction of Aβ peptide with cell membrane. The current study was focused on estimation of cytotoxicity in relation to the intracellular location of the peptide. Thus, we think that our study adds new information regarding the topic. A study that the reviewer suggested was added as a reference to indicate the relationship between cell permeability and cytotoxicity.

2) The Authors hypothesize that the presence of the TAT-tag improves the membrane permeability of the Aβ-peptide and promotes caspase-3/7-like DEVDase activation as well as high cytotoxicity similarly to the untagged Aβ-peptide. Although it is not surprising that this tag improves membrane internalization as already described for other proteins, in my opinion the Authors should better describe the structural effects induced by the tag in Aβ-peptide. In Figure 6 they only compare amyloid fibril formation in the two peptides (tagged and untagged) but they do not provide any information on the intermediates of the aggregation pathway that are responsible for the Aβ-toxicity. In this respect, CD data are not informative as this technique can only read soluble species in solution and other techniques (like FTIR spectroscopy) would be more suitable to monitor different species in solution both soluble and insoluble.

à in the case of tAβ42 the insoluble species are minimal when we check the precipitate after centrifugation. To clarify this, we already commented “This observation was expected because fibrils of tAβ42 were not detected after incubation for up to 48 h (data not shown) (line 469).

3) Most of the experiments in this study have been performed on Hela cells that are not a suitable model for this study. For this reason, the Authors declare (page 5, lane 200 and 201) that the experiment have been performed also in SH-SY5Y5 (I believe there is a mistake as it should be SH-SY5Y), neuroblastoma cell line but they state “SH-SY5Y5 cells showed low apoptotic caspase activation and were easily killed compared with HeLa cells”. In my opinion, results on Aβ-peptide permeability in Hela cell membrane have very low meaning if considering the Alzheimer pathology, above all if they are not reproducible in neuroblastoma cells.

àI am afraid that it is quite difficult to meet the reviewer’s request.  As a part of efforts, we added data obtained by using SH-SY5Y in supplementary figures. 

4) The Authors have also tested the effect of two inhibitors of the Aβ-peptide cytotoxicity, taiwaniaflavone and MBP, on cell internalization for both TAT-tagged and untagged Aβpeptide (Figure 4). They suggest that the protective effect of these inhibitors might be due to their effect on protein internalization. In this respect, the Authors only provide qualitative data for this hypothesis and do not suggest a molecular mechanism involved in the inhibition process (does the inhibitor binds the membrane thus interfering with peptide internalization or the inhibitor is able to bind and stabilize the Aβ-peptide thus avoiding the binding with the cell membrane?). Without providing a molecular hypothesis, in my opinion the interest of these data is very limited for the scientific community.

à The following sentence was added at the end of 3.3 section: “Currently, the molecular mechanism for the inhibitory effect of the above inhibitors are not known. One hypothetical explanation is that the inhibitors potentially block the binding of Aβ peptide to the cell membranes as shown before” with a reference. I am afraid that we were not given enough time to test the hypothesis at this revision period. We plan to investigate regarding it later.

5) Although Aβ-peptide is the most studied amyloid model, References section in this manuscript is very weak and poor and many important papers in this field are completely ignored by the Authors. Clearly, this aspect further weakens this study.

à We updated the references to meet the reviewer’s comment.

Minor issues: Figures: they are not clear, too many labels have been used and this strongly confuse the reader. Also the legends of the figures are difficult to follow. For example, the legend of Figure 4 is fully incomprehensible: it continuously refers to other figures and it is difficult to understand.

à the method of confocal analysis was added instead of referring figure 2. The sentence for method for pre-incubation (The pre-incubation was performed as shown in Figure 3.) was kept, because I think it is better to indicate we used the same method used in figure3. 

Figure 6: CD data should be reported as molar ellipticity ([θ]M) or mean residue ellipticity ([θ]mrw) and not mdeg as they are comparing different proteins (different number of residues).

àchanged as suggested in the figures.

Reviewer 2 Report

Haque et al focused their study on  the cell-permeable peptide TAT-tagged Aβ42 to analyze the implication in cell toxicity . The paper is well written and need one major revision revisions:

  1. Instead of SH-SY5Y cells, should be used HCN-2 cell line that is  a human pediatric cortical neuron culture widely used in in vitro Alzheimer's studies. Is is suggested to repeat the most important experiments in this cell line
  2. Fig 1 columns should be added to be easier to analyze 

Author Response

Reviwer2

1. Instead of SH-SY5Y cells, should be used HCN-2 cell line that is a human pediatric cortical neuron culture widely used in in vitro Alzheimer's studies. It is suggested to repeat the most important experiments in this cell line

à I appreciate your suggestion. However, I am afraid that only 10 days were given for the editing. Thus, this is not possible at the moment. Later we will try this.

2. Fig 1 columns should be added to be easier to analyze 

à I am sorry that I did not understand the comment. Do you mean more labels for each separate figure? The common labels and scales for the figure were omitted. If the current figure is OK, I want keep the format.

Reviewer 3 Report

This is an excellent study that has been carefully curated and reported. The conclusions are well supported by their data, and I am happy to recommend their article for publication following minor changes.

1) Red lines in figure 5 should be removed. 

2) Details of the confocal imaging should be added (resolution, pinhole size, area imaged, etc.). Details of the image processing need to be included, too. Were the images compiled by creating mean Z stacks through the entire cell, or just medial planes, for example?

3) The authors should conduct a secondary analysis of their existing images to see the binding of Ab species to the apical planes of cell membranes. This would help explain why the drugs are blocking Ab internalization, i.e. by potentially blocking the binding of Ab species to the cell membranes in the first place. An example of such an analysis can be found in Limbocker, Nature Communications, 2019 for decoupling Ab internalized in the cell or bound to the cell membrane.

Author Response

1) Red lines in figure 5 should be removed. 

àRemoved

2) Details of the confocal imaging should be added (resolution, pinhole size, area imaged, etc.). Details of the image processing need to be included, too. Were the images compiled by creating mean Z stacks through the entire cell, or just medial planes, for example?

àupdated in Immunocytochemistry section of Materials and Methods.

3) The authors should conduct a secondary analysis of their existing images to see the binding of Ab species to the apical planes of cell membranes. This would help explain why the drugs are blocking Ab internalization, i.e. by potentially blocking the binding of Ab species to the cell membranes in the first place. An example of such an analysis can be found in Limbocker, Nature Communications, 2019 for decoupling Ab internalized in the cell or bound to the cell membrane.

àI am afraid that using our current images, it was difficult to conduct the secondary analysis. We don’t have other images for the analysis. To indicate the importance of the analysis that the reviewer suggested, I put the reference in line 367.

Reviewer 4 Report

The topic of the paper is very actual – it has ben demonstrated that the toxicity of abeta is related to its permeability into cells.

Line 168 fibrillogenesis – the agitation and mixing procedures are very important for fibrillization – it should be specified how the solutions were mixed (for instance for taking aliquots) Why the fibrillization was not monitored continuously in the presence of ThT?

Line 201 – “the SH-SY5Y5 cells showed low apoptotic caspase activation and were easily killed compared with HeLa cells.”  These data must be included at least in the supplement. There are several papers demonstrating that the susceptibility of non-differentiated SH-SH5Y cells towards abeta is relatively low and that caspase activation can also be detected. In principle, the data would be considerably more valuable, if the experiments would be carried out with more proper cellular model of the disease – for instance with differentiated SH-SY5Y..

Line 205 – “Cell death was monitored using the MTT assay“ – the sentence should be correcred – MTT does not show cell death.

Line 459 -  “As the polymerization kinetics of Aβ peptide is not dependent on concentration [13], data obtained for 20 µM concentration of the peptide could be applicable to other concentrations” Can you explain what do you mean? In Ref. 13 Fig 4 there is a clear dependence of the fibrillzation rate on abeta concentration and there are a number of more recent papers where the dependence is studied extensively.

Fig. 6 panel A – no lab period characteristic to fibrillization was observed and a considerable fluorescence intensity is present at time 0. Does this mean that your peptide is not completely unfibrillizedand contain fibrils? If you repeat the HFIP treatment two or three times would the lag-period characteristic to fully monomerized peptide appear? 

The analysis of literature is rather outdated – within the references are only three papers from the last   six years and all three are co-authored by the authors of manuscript.  The result should be put into context with latest data from the field.

Minor remarks –

  1. line 27 – amyloid peptide is defined as a „group of peptides with length 36-43“, however, it is not defined in this way in the references included. This remark is not crucial considering the topic of the paper, however it must be corrected.

Author Response

Line 168 fibrillogenesis – the agitation and mixing procedures are very important for fibrillization – it should be specified how the solutions were mixed (for instance for taking aliquots) Why the fibrillization was not monitored continuously in the presence of ThT?

à It has been edited (fibrillogenesis section of materials and methods)

Line 201 – “the SH-SY5Y5 cells showed low apoptotic caspase activation and were easily killed compared with HeLa cells.”  These data must be included at least in the supplement. There are several papers demonstrating that the susceptibility of non-differentiated SH-SH5Y cells towards abeta is relatively low and that caspase activation can also be detected. In principle, the data would be considerably more valuable, if the experiments would be carried out with more proper cellular model of the disease – for instance with differentiated SH-SY5Y.

à Supplementary data of SH-SH5Y cells were added.

Line 205 – “Cell death was monitored using the MTT assay “– the sentence should be correcred – MTT does not show cell death.

à Changed to “Cell viablility was assessed by the MTT assay”

Line 465 - “As the polymerization kinetics of Aβ peptide is not dependent on concentration [13], data obtained for 20 µM concentration of the peptide could be applicable to other concentrations” Can you explain what do you mean? In Ref. 13 Fig 4 there is a clear dependence of the fibrillzation rate on abeta concentration and there are a number of more recent papers where the dependence is studied extensively.

à In ref.13 fig4A showed the initial rate at the difference concentrations. The initial rate [F(t)/min] should be higher with the higher concentration of the peptide. But t1/2 (half time for completion of the polymerization) is a real parameter explaining the kinetics. t1/2 is the time when Log{F(t)/[F(∞)-F(t)]}=0 and is independent on the concentration. F(t) and F(∞) is fluorescence at time t and ∞. A reference (Biochemistry 37:17882) was added to clarify this.

Fig. 6 panel A – no lab period characteristic to fibrillization was observed and a considerable fluorescence intensity is present at time 0. Does this mean that your peptide is not completely unfibrillizedand contain fibrils? If you repeat the HFIP treatment two or three times would the lag-period characteristic to fully monomerized peptide appear? 

à I guess the reviewer meant lag period. In the figure we wanted to show the long incubation time (up to 48 h) to show the poor fibrillogenensis of tAβ42 and the lag phase is too short to be seen in the figure. I did not edit this, because the data of long incubation samples is more important than showing the lag phase.

The analysis of literature is rather outdated – within the references are only three papers from the last   six years and all three are co-authored by the authors of manuscript.  The result should be put into context with latest data from the field.

à references were update

Minor remarks –

  1. line 27 – amyloid peptide is defined as a „group of peptides with length 36-43“, however, it is not defined in this way in the references included. This remark is not crucial considering the topic of the paper, however it must be corrected.

à Edited and updated

Round 2

Reviewer 1 Report

The Authors have answered most of my concerns and, in my opinion, this manuscript is much improved upon suggestion of all reviewers. For these reasons, I believe this revised version can be accepted for publication in Life.

Reviewer 2 Report

Dear, I think that only 10 days were not sufficient to improve the parer and
others reviewer were more hard then me, so I think that the paper nedd more changes before publishing